# Improving Subgroup Robustness via Data Selection

**Saachi Jain**,* **Kimia Hamidieh**,* **Kristian Georgiev**,*
Andrew Ilyas, Marzyeh Ghassemi, Aleksander Mądry
MIT
`{saachij,hamidieh,krisgrg,ailyas,mghassem,madry}@mit.edu`

## Abstract

Machine learning models can often fail on subgroups that are underrepresented during training. While dataset balancing can improve performance on underperforming groups, it requires access to training group annotations and can end up removing large portions of the dataset. In this paper, we introduce *Data Debiasing with Datamodels* (D3M), a debiasing approach which isolates and removes specific training examples that drive the model's failures on minority groups. Our approach enables us to efficiently train debiased classifiers while removing only a small number of examples, and does not require training group annotations or additional hyperparameter tuning.

## 1 Introduction

The advent of large datasets such as OpenImages [23] and The Pile [13] has led to machine learning models being trained on explicit [4] and illegal [48] content, or on data that encode negative societal biases [6, 12, 2, 8] and other spurious correlations [32, 30]. On one hand, there is increasing evidence that models reflect the biases in these datasets; on the other hand, the enormous scale of these datasets makes it extremely expensive to manually curate them—and so removal of such "harmful data" is challenging.

In this paper, we propose an approach that aims to remove data responsible for biased model predictions. In particular, we focus on a specific way of quantifying model bias— *worst-group error*— which captures the extent to which model performance degrades on pre-defined subpopulations of the data. We aim to identify (and remove) the points in the training dataset that contribute most to this metric to improve the model's group robustness.

The challenge inherent in this approach is that it requires an understanding of how training data affect machine learning model predictions. To overcome this challenge, we first approximate predictions as simple, direct functions of the training dataset, using a framework called *datamodeling* [18, 33]. We can then write our quantitative notion of model bias (which is a function of predictions) as a function of the dataset. Finally, by studying this function, we identify the training data points that contribute most to this measure of model bias. With the resulting method, which we call *Data Debiasing with Datamodels* (D3M), we show that, across a variety of datasets, there are often a small number of examples that disproportionately drive worst-group error. Removing these examples, in turn, greatly improves models' worst-group error while maintaining dataset size.

**Roadmap & contributions.** In the rest of this paper, we present and demonstrate the effectiveness of our Data Debiasing with Datamodels (D3M). Concretely, we show that D3M enables us to:

- **Pinpoint examples that harm worst-group accuracy.** We show that there are often a small number of examples that disproportionately drive models' worst-group error on validation data. For example, on `CelebA-Age`, our method improves worst group error over

38th Conference on Neural Information Processing Systems (NeurIPS 2024).

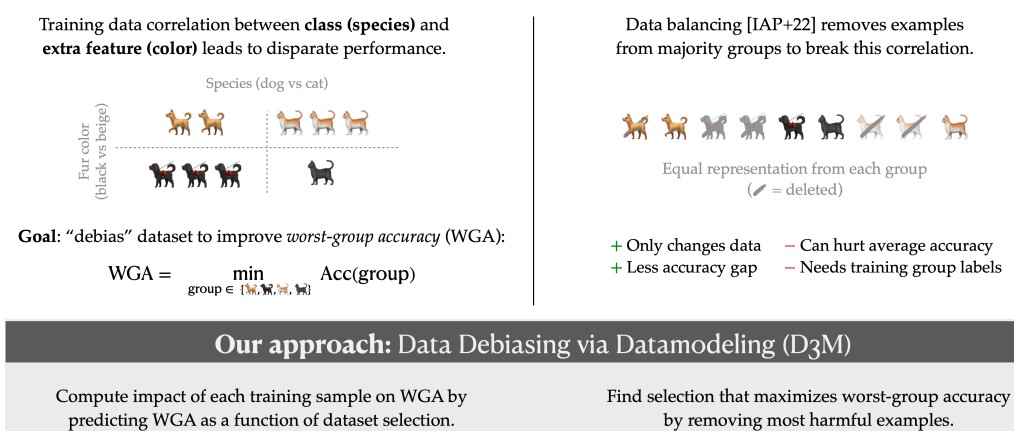

Figure 1: Our method (D3M) improves worst group accuracy by identifying and removing the training samples which most negatively impact worst-group accuracy. Specifically, we use TRAK [33] to identify examples that exacerbate the discrepancy in group performance. We then remove and re-train a model on the remaining data.

a natural baseline (data balancing) while removing $2.4\times$ fewer examples. Furthermore, these offending examples often form coherent subpopulations within the data.

- **Achieve competitive debiasing performance.** Our approach outperforms standard approaches (both model-based and data-based) to improving worst-group accuracy [25, 21, 17], and is able to match the performance of methods which use ground-truth training group annotations [40].

- **Discover unlabeled biases.** When validation group labels are unavailable, we show how to extract hidden biases (i.e., unlabeled subgroups) directly from the data. As a result, we can perform end-to-end debiasing without *any* group annotations.

We present our method in Section 4, and demonstrate these capabilities in Section 5. In Section 6, we leverage our framework to discover and mitigate biases within the ImageNet dataset, where D3M surfaces coherent color and co-occurrence biases. We then debias the model according to these failures, and improve accuracy on the identified populations.

## 2 The group robustness problem

We consider an (unobserved) data distribution $\mathcal{D}$ over triplets $(x_i, y_i, g_i)$, each comprising an input $x_i \in \mathcal{X}$, a label $y_i \in \mathcal{Y}$, and a *subgroup label* $g_i \in \mathcal{G}$, where $\mathcal{G}$ is the set of distinct subpopulations in the data. As a running example, consider the CelebA age classification task—here, we take the inputs $x_i$ to be images of faces, the labels $y_i$ to be either "old" or "young," and the possible group labels to be "old man", "old woman", "young man", and "young woman" (see Figure 1).

Given a training dataset $S_{\text{train}}$ and a (small) validation dataset $S_{\text{val}}$, the goal of the group robustness problem is to produce a classifier $f$ that minimizes the worst-case loss over groups, i.e.,

$$\max_{g' \in \mathcal{G}} \mathbb{E}_{(x,y,g)\sim\mathcal{D}} \left[ \ell(f(x), y) \big| g = g' \right], \tag{1}$$

where $\ell(\cdot, \cdot)$ is a loss function. When $\ell$ is the 0-1 loss, Equation (1) is (one minus) the *worst-group accuracy* (WGA) of the classifier $f$, which we use to quantify success in the remainder of this work.

Standard loss minimization can yield models that perform poorly with respect to (1). For instance, returning to our example of CelebA age classification, suppose there was a spurious correlation between age and gender in the training set $S_{\text{train}}$, such that old men and young women are overrepresented. A predictor that minimizes loss on $S_{\text{train}}$ might leverage this correlation, and thus perform poorly on the underrepresented subgroups of old women or young men.

In practice, subgroup labels $g_i$ can be expensive to collect. Thus, approaches to the subgroup robustness problem vary in terms of whether we observe the group label $g_i$ in the training set $S_{\text{train}}$ and in the validation set $S_{\text{val}}$. In particular, there are three settings of interest:

- **Full-information (Train ✓/ Val ✓):** We observe the group labels for both the training dataset $S_{\text{train}}$ and validation dataset set $S_{\text{val}}$.

- **Partial-information (Train ✗/ Val ✓):** We observe the group labels for the validation set $S_{\text{val}}$, but not for the (much larger) training set $S_{\text{train}}$.

- **No-information (Train ✗/ Val ✗):** We do not have group information for either $S_{train}$ or $S_{\text{val}}$. Note that theoretically this setting is unsolvable, since for any non-perfect classifier $f$, there exists an assignment of group labels so that the worst-group accuracy is zero. Nevertheless, subgroups of relevant practical interest typically have structure that allows for non-trivial results even with no information.

In this work, we focus on the partial-information and no-information settings, since acquiring group labels for the entire training set is often prohibitively expensive. Still, in Section 5, we show that our proposed methods (D3M for the partial-information setting, and AUTO-D3M for the no-information setting) perform comparably to full-information approaches.

## 3 Related work

Before introducing our method, we discuss a few related lines of work.

**Approaches to subgroup robustness.** The *group robustness* problem (Section 2) has attracted a wide variety of solutions (see, e.g., [3, 20, 40, 25, 21, 38]). Broadly, these solutions fall into one of two categories—model interventions and data interventions. *Model interventions* target either model weights [42, 45] or the training procedure [40, 21]. *Data interventions*, on the other hand, seek to improve worst-group accuracy by modifying the training dataset. For example, data balancing removes or subsamples examples so that all subgroups are equally represented. Idrissi et al. [17] find that this simple approach can performs on par with much more intricate model intervention methods.

In this work, we focus on data interventions, for two reasons. First, it is often training data that drives models' disparate performance across groups [27], e.g., via spurious correlations [32] or underrepresentation [6]. Second, data interventions do not require any control over the model training procedure, which can make them a more practical solution (e.g., when using ML-as-a-service). Indeed, since data intervention approaches only manipulate the dataset, they are also easy to combine with model intervention techniques.

Compared to our work, the main drawback of existing data interventions is that they often (a) require subgroup labels for the training data (which might not be available), and (b) hurt the models' natural accuracy on skewed datasets [7, 44]. In this work we circumvent these limitations, by proposing a data-based approach to debiasing that can preserve natural accuracy without access to subgroup information.

**Bias discovery.** Another related line of work identifies biases in machine learning datasets and algorithms. For the former, previous works have shown that large, uncurated datasets used for training machine learning models often contain problematic or biased data [4, 5, 48]. Raji et al. [39] show that data bias can be a hurdle towards deploying functional machine learning models. Nadeem et al. [29] curate a dataset to estimate bias in NLP models. Adebayo et al. [1] show that label errors can disproportionately affect disparity metrics.

On the learning algorithm side, Shah et al. [47] and Puli et al. [37] show that the inductive bias of neural networks may encourage reliance on spurious correlations. Pezeshki et al. [34] leverage two networks trained on random splits of data while imitating confident held-out mistakes made by its

sibling to identify the bias. Shah et al. [46] show that algorithmic design choices (e.g., the choice of data augmentation) can significantly impact models' reliance on spurious correlations. Finally, there has been a variety of work on "slice discovery" [19, 10], where the goal is to discover systematic errors made by machine learning models.

**Data selection for machine learning.** Our work uses data selection to improve subgroup robustness of machine learning models. A recent line of work has explored data selection for improving various measures of model performance. For example, Engstrom et al. [9] leverage datamodeling [18, 33] to select pretraining data for LLMs. Similarly, Xia et al. [52] and Nguyen and Wong [31] select data for finetuning, and in-context learning, respectively. In another related work, Wang et al. [49] propose a method to reweight training data in order to improve models' fairness.

Many of these works leverage *contributive data attribution* methods to select data that improves model performance [51]. Koh and Liang [22], Feldman and Zhang [11], Schioppa et al. [43], and Hammoudeh and Lowd [15] propose using different variants of influence functions. Ghorbani and Zou [14] leverage connections to Shapley values, a concept from game theory. Pruthi et al. [36] use a heuristic to estimate the contribution of each data point to model performance. Ilyas et al. [18] and Park et al. [33] use the datamodeling framework.

# 4 Debiasing datasets with datamodeling (D3M)

In this section, we present our *data-based* approach to training debiased classifiers. The main idea behind our approach is to identify (and remove) the training samples that negatively contribute to the model's worst-group accuracy, by writing model predictions as a function of the training data.

**Preliminaries.** Let $S = \{(x_1, y_1), \ldots, (x_n, y_n)\}$ be a dataset of input-label pairs. For any subset of the dataset—as represented by indices $D \subset [n]$—let $\theta(D) \in \mathbb{R}^p$ be the parameters of a classifier trained on $D$. Given an example $z = (x, y)$, let $f(z; \theta)$ be the correct-class margin on $z$ of a classifier with parameters $\theta$ (defined as $\log(\frac{p}{1-p})$, where $p$ is the confidence assigned to class $y$ for input $x$).

A *datamodel* for the example $z$ is a simple function that predicts $f(z; \theta(D))$ directly as a function of $D$, i.e., a function $\hat{f}_z : 2^{[n]} \to [0, 1]$ such that

$$\hat{f}_z(D) \approx f(z; \theta(D)) \qquad \text{for } D \subset [n].$$

Recent works (e.g., [18, 24, 33]) demonstrate the existence of accurate *linear* datamodels—functions $\hat{p}$ that decompose *additively* in terms of their inputs $D$. In other words, these works show that one can compute example-specific vectors $\tau(z) \in \mathbb{R}^n$ such that

$$\hat{f}_z(D) := \sum_{i \in D} \tau(z)_i \approx f(z; \theta(D)). \tag{2}$$

The coefficients $\tau(z)_i$ have a convenient interpretation as quantifying the "importance" of the $i$-th training sample to performance on example $z$ (i.e., as a *data attribution* score [15]). In what follows, we will assume access to coefficients $\tau(z)$ for any example $z$—at the end of this section, we will show how to actually estimate the coefficient vectors $\tau(z)$ efficiently.

**Debiasing approach.** How can we leverage datamodeling to debias a dataset? Recall that our goal is to remove the samples in $S$ that lead to high worst-group error. Stated differently, given a dataset $S$ of size $n$, we want to maximize the worst-group performance of a classifier $\theta(D)$ with respect to the indices $D \subset [n]$ that we train on.

Our main idea will be to approximate the predictions of $\theta(D)$ using the corresponding datamodels $\hat{f}_z(D)$. To illustrate this idea, suppose that our goal was to maximize performance on a *single* test example $z$, i.e., $\arg\max_D f(z; \theta(D))$. We can approximate this goal as finding $\arg\max_D \hat{f}_z(\theta(D))$: then, due to the linearity of the datamodel $\hat{f}_z$, the training samples that hurt performance on $z$ are simply the bottom indices of the vector $\tau(z)$.

Now, this analysis applies not only to a single example $z$, but to any *linear combination* of test examples. In particular, if we wish to maximize performance on a linear combination of validation examples, we simply take the linear combination of their coefficients, and remove the training examples corresponding to the smallest coordinates of the averaged vector.

**Debiasing with group-annotated validation data.** Given a set of validation samples for which the group labels $g_i$ are observable, our last observation gives rise to the following simple procedure:

1. **Compute group coefficients $\tau(G)$ for each $G$.** Since we have group annotations for each validation sample, we can define a vector $\tau(G)$ for each group $G \in \mathcal{G}$ as simply the average $\tau(z)$ within each group.

2. **Compute group alignment.** Next, we compute a *group alignment score* $A_i$ for each training sample $i \in [n]$, which captures the the impact of the sample on worst-group performance. Since there may be many low-performing groups, we use a "smooth maximum" function to weight each group according to its average loss. Thus, for a training example $i$,

$$A_i = \frac{\sum_{g \in \mathcal{G}} \exp(\beta \ell_g) \cdot \tau(g)_i}{\sum_{g' \in \mathcal{G}} \exp(\beta \ell_{g'})}, \text{ where we set hyperparameter } \beta = 1. \qquad (3)$$

   Here, $\ell_g$ is the loss of a base classifier $\theta(S)$ on group $g$ (evaluated on the given validation set).

3. **Remove drivers of bias.** Finally, we construct a new training set $S_{\text{new}}$ by keeping only the examples with the highest group alignment scores, i.e., removing the examples that most degrade worst-group accuracy:

$$S_{\text{new}} = \arg \text{top-k}(\{A_i : z_i \in S_{\text{train}}\}).$$

We make two brief observations about hyperparameters before continuing. When computing the group alignment score in Step 2, the hyperparameter $\beta$ controls the temperature of the soft maximum function in (3). When $\beta \to 0$, the group alignment $A_i$ measures the impact of the $i$-th training example on the "balanced" performance (treating all groups equally). As $\beta \to \infty$, $A_i$ collapses to the training example's importance to *only the worst group*, which is suboptimal if models perform poorly on more than one group. For simplicity, we take $\beta = 1$ and refrain from tuning it.

Another hyperparameter in the algorithm above is the number of examples to remove, $k$. We consider two different ways of setting this hyperparameter. One approach is to search for the value of $k$ that maximizes worst-group accuracy on the validation set $S_{\text{val}}$. Alternatively, we find that the simple (and much more efficient) heuristic of removing all examples with a negative group alignment score (i.e., examples for which $A_i < 0$) tends to only slightly over-estimate the best number of examples to remove (see, e.g., Figure 2). Thus, unless otherwise stated, we use this heuristic when reporting our results.

**Debiasing *without* group annotations.** Our procedure above relies on group annotations for a validation set $S_{\text{val}}$ to compute the "per-group coefficients" $\tau(G)$. In many real-world settings, however, models might exhibit disparate performance along *unannotated* subpopulations—in this case, we might not have a validation set on which we can observe group annotations $g_i$. Can we still fix disparate model performance in this setting?

In general, of course, the answer to this question is no: one can imagine a case where each individual example is its own subpopulation, in which case worst-group accuracy will be zero unless the classifier is perfect. In practical settings, however, we typically care about the model's disparate performance on coherent groups of test examples. The question, then, becomes how to find such coherent groups.

We posit that a unifying feature of these subpopulations is that they are *data-isolated*, i.e., that models' predictions on these coherent groups rely on a different set of training examples than models' predictions on the rest of the test data. Conveniently, prior works [18, 46] show that to find data-isolated subpopulations, one can leverage the *datamodel matrix*—a matrix constructed by stacking the datamodel vectors $\tau(z)$ for each test example. Intuitively, the top principal component of this matrix encodes the direction of maximum variability among the vectors $\tau(z)$. Thus, by projecting the datamodel vectors $\tau(z)$ of our validation examples onto this top principal component, we can identify the examples that are, in a sense, "maximally different" from the rest of the test examples in terms of how they rely on the training set. These maximally different examples correspond to an isolated subpopulation, to which we can apply D3M directly.

This approach (which we call AUTO-D3M), enables us to perform end-to-end debiasing without *any* group annotations. This method proceeds in four steps. For each class:

1. Construct a matrix $\mathbf{T}$ of stacked attribution vectors, where $\mathbf{T}_{ij} = \tau(z_i)_j$.
2. Let $\boldsymbol{v}$ be the top principal component of $\mathbf{T}$.
3. Project the attribution vector $\tau(z)$ onto $\boldsymbol{v}$ and construct "group pseudo-labels"
$$g_i = \mathbf{1}\{\tau(z_i)^\top \boldsymbol{v} \geq \lambda\}.$$
where $\lambda$ is a hyperparameter [2]
4. Apply D3M with the group pseudo-labels to train a debiased classifier.

**Estimating the coefficients $\tau(z)$.** In order to operationalize D3M and AUTO-D3M, it remains to show that we can actually estimate coefficients $\tau(z)$ satisfying (2). To accomplish this, we use a method called TRAK [33]. Leveraging differentiability of the model output $f(z; \theta)$ with respect to the model parameters $\theta$, TRAK computes the coefficient vector $\tau(z)$ for an example $z$ as follows:

(a) Train a model $\theta^* := \theta(S)$ on the entire training dataset $S = \{z_1, \ldots, z_n\}$.
(b) Sample a random Gaussian matrix $\mathbf{P} \in \mathbb{R}^{p \times k}$ where $p$ is the dimensionality of $\theta^*$ (i.e., the number of model parameters) and $k$ is a hyperparameter;
(c) For an example $z$, define $g(z) := \mathbf{P}^\top \nabla_\theta f(z; \theta^*)$ as the randomly-projected model output gradient (with respect to the model parameters) evaluated at $z$.
(d) Compute the coefficient vector
$$\underbrace{\tau(z)_i}_{i\text{-th coefficient for example } z} = g(z)^\top \left( \sum_{z_j \in S} g(z_j) \cdot g(z_j)^\top \right)^{-1} g(z_i) \cdot (1 - \sigma(f(z; \theta^*)))$$
(e) Repeat steps (a)-(d) for $T$ trials, and average the results to get a final coefficient vector $\tau(z)$. The trials are identical save for the randomness involved in step (a).

We provide more intuition and details behind TRAK in Appendix A.

**A note on scalability** In terms of computational cost, TRAK involves taking a single backward pass on each of the training and validation examples to compute the model's gradient. The (projected) gradients are then saved to compute TRAK scores. Typically, TRAK is computed over $T$ trials: following the original paper we use $T = 100$. However, our approach can be used with any datamodeling technique.

## 5 Results

In Section 4, we presented D3M—an approach for debiasing a classifier by identifying examples which contribute to a targeted bias. In this section, we validate this framework by assessing its performance on tasks with known biases.

We consider four classification tasks where there is a spurious correlation between the target label and a group label in the training dataset: `CelebA-Age` [26, 19], `CelebA-Blond` [26], `Waterbirds` [41], and `MultiNLI` [50]. We provide more information about the datasets in Appendix B.1, and other experimental details in Appendix B.2.

### 5.1 Quantitative results

We first evaluate D3M and AUTO-D3M quantitatively, by measuring the worst-group accuracy of models trained on the selected subsets of the biased datasets above.

**D3M: Debiasing the model in the presence of validation group labels.** In Table 1, we compare D3M against several baselines, each of which requires either only validation group labels (✗/ ✓) or both training and validation group labels (✓/ ✓). We find that D3M outperforms all other methods that use the same group information (i.e., only validation group labels) on all datasets except Waterbirds[3].

---

[2]For our experiments we choose $\lambda$ so that the lower performing group consists of 35% of the validation examples of that class.

[3]Note that WaterBirds has more worst-group examples in the `val` split (133) than the `train` split (56). Since DFR directly fine-tunes on the validation set, it has an advantage here over other methods.

| Group Info Train / Val | Method | Worst Group Accuracy (%) | | | |
|---|---|---|---|---|---|
| | | CelebA-Age | CelebA-Blond | Waterbirds | MultiNLI |
| ✗/ ✗ | ERM | 56.7 | 45.9 | 57.9 | 67.2 |
| | **AUTO-D3M (ours)** | **76.0** | **83.8** | **81.0** | **75.0** |
| ✗/ ✓ | JTT [25] | 61.0 | 81.6 | 63.6 | 72.6 |
| | DFR* [21] | 70.4 | 88.4 | **89.0** | 74.7 |
| | **D3M (ours)** | **75.6** | **90.0** | 87.2 | **76.0** |
| ✓/ ✓ | RWG [17] | **75.6** | 88.4 | 81.2 | 68.4 |
| | SUBG [17] | 68.5 | 88.3 | **85.5** | 67.8 |
| | GroupDRO [40] | 74.8 | **90.6** | 72.5 | **77.7** |

Table 1: Worst-group accuracies on four group robustness datasets. A $*$ denotes methods that use validation group labels for both finetuning and hyperparameter tuning.

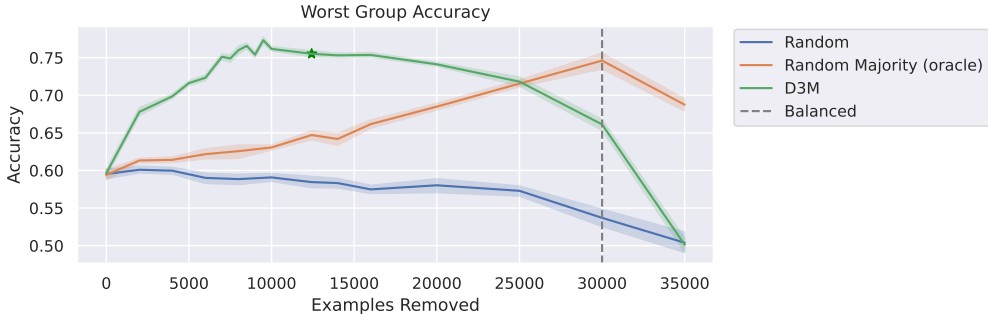

Figure 2: Worst group accuracy on `CelebA-Age` as a function of the number of examples $k$ removed from the training set, using various removal methods. In green, D3M removes the $k$ training examples with the most negative alignment scores $A_i$. The green star marks the value of $k$ selected by our heuristic ($A_i < 0$). In blue is the performance of a random baseline that removes $k$ examples at random from the training set, and in orange is dataset balancing, which removes examples randomly from the majority group. Compared to baselines, D3M efficiently improves worst group accuracy.

Moreover, D3M performs on par with methods that have full access to both training and validation group labels.

**AUTO-D3M: Discovering biases with TRAK.** We now consider the case where validation group labels are not accessible. Using AUTO-D3M, we debias our model using pseudo-annotations derived from the top principal component of the TRAK matrix (AUTO-D3M in Table 1)[4]. Note that AUTO-D3M is the only method other than ERM that does not require either train or validation group labels. Despite this, AUTO-D3M achieves competitive worst-group accuracy in our experiments. We emphasize that AUTO-D3M does not require group labels at all—in particular, we *do not* use group labels to do hyperparameter selection or model selection when we retrain.

**The effect of the number of removed examples $k$.** How well does D3M isolate the training examples that drive disparate performance? To answer this question, we iteratively remove training examples from `CelebA-Age` starting with the most negative $A_i$ and measure the worst-group and balanced accuracy (See Figure 2). `CelebA-Age` has 40K "majority" examples and 10K "minority" examples; thus, naive balancing requires removing 30K training examples. In contrast, by isolating *which* specific majority examples contribute to the bias, our method is able to debias the classifier by removing only 10K examples.

Our heuristic of removing examples with negative $A_i$ (the star in Figure 2) slightly overestimates the best number of examples to remove. Thus, while this heuristic gives a decent starting point for $k$, actually searching for the best $k$ might further improve performance.

---

[4]For `MultiNLI`, we chose the PCA component by inspection that captures examples with/without negation.

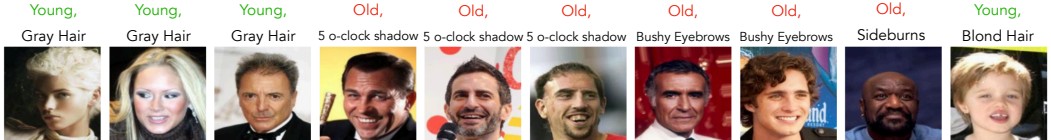

Figure 3: Randomly sampled examples from the subpopulations with the most negative group alignment scores. We find that many of these examples have labeling errors (e.g., platinum blond instead of gray hair.)

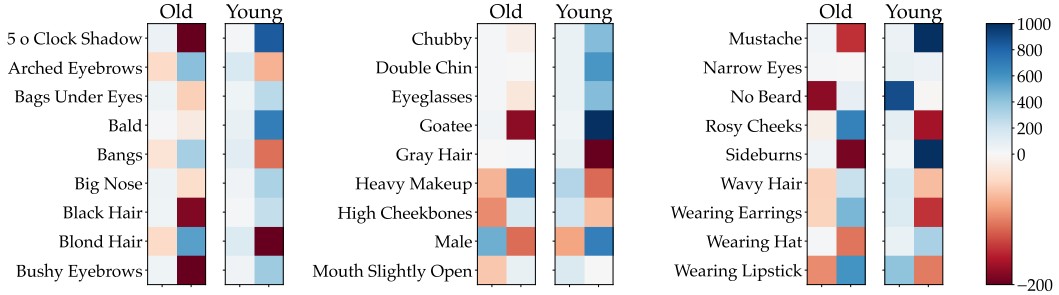

Figure 4: The average group alignment score of the training examples in each subpopulation of `CelebA-Age`. Subpopulations such as "old" with "bushy eyebrows" or "young" with "gray hair" have particularly negative scores.

## 5.2 Qualitative results

What type of data does our method flag? In particular, do the examples we identify as driving the targeted bias share some common characteristics? To test this hypothesis, we inspect the data removed by our method and identify subpopulations within the majority groups that are disproportionately responsible for the bias. We then retrain the model after excluding *all* training examples from the identified subpopulations and show that this is a viable strategy for mitigating the underlying bias.

**Finding subpopulations responsible for model bias.** Consider the running example from Figure 1 where we train a model on the `CelebA-Age` dataset to predict whether a person is "young" or "old" when gender is a spurious feature (such that young women and old men are overrepresented). `CelebA-Age` has a variety of annotations beyond age and gender, such as whether the person is wearing eyeglasses. In this section, we use these extra annotations to identify coherent subpopulations that are flagged by our methods.

In particular, we consider subpopulations formed by taking the Cartesian product of labels and annotations, e.g., subpopulations of the form ("young", "wearing eyeglasses"). For each of these subpopulations, we calculate the average group alignment score $A_i$ of the training examples within that subpopulation (see Figure 4). We find that subpopulations such as "young" with "gray hair" or "old" with either "5 o'clock shadow" or "busy eyebrows" have particularly negative group alignment scores. In Figure 3, we show examples from the subpopulations with the most negative group alignment scores, and observe that a large fraction of the examples in these subpopulations contain labeling errors.

**Retraining without identified subpopulations.** Once we have identified subpopulations with negative alignment scores, a natural strategy for mitigating the underlying bias is to exclude these subpopulations from the training set. To explore this approach, we exclude the five subpopulations with the most negative attribution scores on average from the `CelebA-Age` dataset: "Young" + "Gray Hair", "Old"+ "5 o'Clock Shadow", "Old" + "Bushy Eyebrows", "Young" + "Blond Hair", and "Old" + "Sideburns." After retraining the model on this modified training set, we get a worst-group accuracy (WGA) of $68.4\%$—an improvement of ~12% over the WGA of the original model ($56.7\%$).

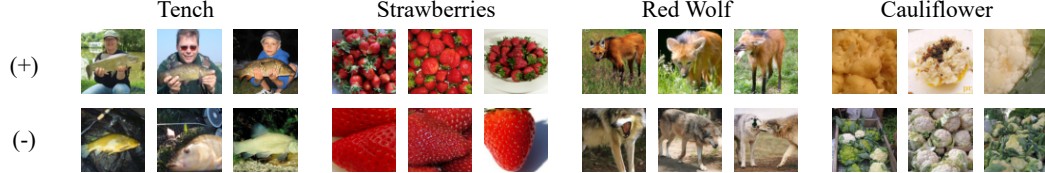

Figure 5: For four ImageNet classes, the most extreme (positive or negative) examples according to the top PCA direction of the TRAK matrix. Our method identifies color and co-occurrence biases.

## 6 Case Study: Finding and Mitigating Model Failures on ImageNet

In Section 5 we evaluated D3M and AUTO-D3M on datasets where the bias was already known. We now deploy AUTO-D3M to discover and mitigate biases within the ImageNet dataset, which does not have a predefined bias or available group annotations.

**Identifying ImageNet biases.** We use TRAK to compute a coefficient matrix $\mathbf{T}$ (see Step 1 of AUTO-D3M in Section 4) for a held out validation split (10% of the training set). Focusing on seven ImageNet classes, we use the first principal component of the matrix $\mathbf{T}$ to identify potential biases. In Figure 5, we display the most extreme training examples according to the top principal component for four of these classes. PCA identifies semantically color and co-occurrence biases (e.g., tench fishes with or without humans or yellow/white cauliflowers that are either cooked or uncooked.) In fact, our identified biases match the challenging subpopulations in Jain et al. [19] and Moayeri et al. [28].

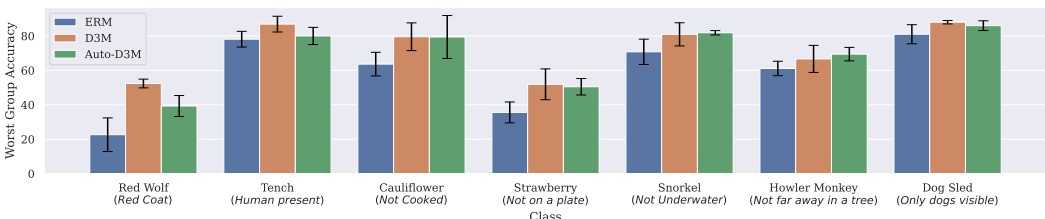

Figure 6: Worst-group accuracy for the ImageNet classes studied in Section 6 after intervening with either D3M or AUTO-D3M (error bars over 1 std).

**Mitigating ImageNet biases with AUTO-D3M.** For each of the four targeted ImageNet classes, we seek to mitigate the identified failure modes with AUTO-D3M. We consider two settings based on the level of human intervention. In the first, we manually assign each of the validation images to a group according to a human description of identified bias (e.g., an image of a tench is in group 1 if a human is present and group 2 otherwise), and then use those group labels with D3M. [5] In the second setting, we debias in a purely automatic fashion, using AUTO-D3M to derive pseudo-group labels from the top principal component. In Figure 6, we display worst group accuracy on the test images of the targeted class (evaluated using manual group assignments of the 50 test examples). Both D3M and AUTO-D3M improve worst group accuracy over ERM without significantly impacting the overall ImageNet accuracy (see Appendix C.2).

## 7 Conclusion

We propose Data Debiasing with Datamodels (D3M), a simple method for debiasing classifiers by isolating training data that disproportionately contributes to model performance on underperforming groups. Unlike approaches such as balancing, our method only removes a small number of examples and does not require training group annotations or additional hyperparameter tuning. More generally, our work takes a first step towards *data-centric* model debiasing.

---

[5]Here, we only consider the target class when computing the loss weighting. As a result, the heuristic overestimates the number of examples $k$ to remove, and so we instead search for the optimal $k$ using our held out validation set.

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

# A  Background on Data Attribution and TRAK

Let $\mathcal{Z}$ be the input space and $\mathcal{S}$ be a training set of interest. For a given training subset $S' \subset S$ and target example $z$, data attribution techniques seek to estimate the attribution score of $z$ — that is, the change in the model's prediction on $z$ when the model is trained on the subset $S'$. More formally, let $f(z, \theta(S))$ be the model's output function on example $z$. Then we can define $\tau(z)_i$ as the attribution score of the $i$th training example $z_i$ on target example $z$

$$\tau(z)_i = f(z, \theta(S)) - f(z, \theta(S \backslash z_i)).$$

While $\tau(z)$ is relatively straightforward to compute for linear models [35], computing this influence is far more challenging for neural networks. Thus, in order to approach this problem, TRAK first approximates $f(z; \theta(S))$ as a linear model on top of the gradients $\nabla_\theta(f; \theta^*)$ of the original neural network after convergence. We can then plug this approximation into the estimate for linear classifiers to approximate $\tau(z)$. After simplification, the TRAK estimate of the influence of $z$ is

$$\tau(z) = -\phi(z)^T (\Phi^T \Phi)^{-1} \Phi \mathbf{Q}$$

where $\phi(z) = \nabla f(z; \theta(S^*))$ are the (randomly projected) gradients on example $z$, $\Phi$ are the stacked training gradients $\Phi = [\phi(z_1), ..., \phi(z_n)]$, and $\mathbf{Q}$ is a normalization matrix.

# B  Details of Experiments

## B.1  Experimental Setup

In this section, we describe the datasets, models and evaluation procedure that we use throughout the paper.

**Datasets.**   In order to cover a broad range of practical scenarios, we consider the following image classification and text classification problems.

- `Waterbirds` [40] is a binary image classification problem, where the class corresponds to the type of the bird (landbird or waterbird), and the background is spuriously correlated with the class. Namely, most landbirds are shown on land, and most waterbirds are shown over water.

- `CelebA-Blond` [26] is a binary image classification problem, where the goal is to predict whether a person shown in the image is blond; the gender of the person serves as a spurious feature, as 94% of the images with the "blond" label depict females.

- `CelebA-Age` [26, 19] is a binary image classification problem, where the goal is to predict whether a person shown in the image is young; the gender of the person serves as a spurious feature. For this task, we specifically subsample the training set such that the ratio of samples in the majority vs. minority groups is 4:1.

- `MultiNLI` [50, 40] is a classification problem where given a pair of sentences, the task is to classify whether the second sentence is entailed by, neural with, or contradicts the first sentence. The spurious attribute from Sagawa et al. [40] describes the presence of negation words, which appear more frequently in the examples from the negation class.

**Methods.**   We benchmark our approach against the following methods:

- **ERM** is simple empirical risk minimization on the full training set.

- **RWG** [17] is ERM applied to random batches of the data where the groups are equally represented with a combination of upsamping and downsampling such that the size of the dataset does not change.

- **SUBG** [17] is ERM applied to a random subset of the data where we subsample all groups such that they have the same number of examples.

- **GroupDRO** [40] trains that minimizes the worst-case performance over pre-defined groups in the test dataset.

- **Just Train Twice (JTT)** [25] trains an ERM model with upsamping initially misclassified training examples by an initial ERM model.
- **DFR** [21] trains an ensemple of linear models on a balanced validation set, given ERM features.

## B.2   Training Details

In this section, we detail the model architectures and hyperparameters used by each approach. We used the same model architecture across all approaches: Randomly initialized ResNet-18 [16] for CelebA and ImageNet-pretrained ResNet-18s for Waterbirds. We use the GroupDRO implementation by Sagawa et al. [40] and DFR implementation by Kirichenko et al. [21].

For all approaches, we tune hyperparameters for ERM-based methods (ERM, DFR, and D3M) and re-weighting based methods (RWG, SUBG, GroupDRO and JTT) separately. For RWG, SUBG, GroupDRO and JTT, we early stop based on highest worst-group accuracy on the validation set as well. We optimize all approaches with Adam optimizer.

For the CelebA dataset, we all methods with learning rate $1e - 3$, weight decay $1e - 4$, and batch size 512. We train RWG, SUBG, GroupDRO and JTT with learning rate $1e - 3$, weight decay $1e - 4$, and batch size 512. We train all models for the `CelebA-Age` task to up to 5 epochs and all models for `CelebA-Blond` task up to 10 epochs.

For the Waterbirds dataset, we train the approaches that use the ERM objective (including D3M) with learning rate $1e - 4$, weight decay $1e - 4$, and batch size 32. We train RWG, SUBG, GroupDRO and JTT with learning rate $1e - 5$, weight decay $0.1$, and batch size 32. We train all models to up to 20 epochs.

For all other hyperparameters, we use the same hyperparameters as Kirichenko et al. [21] for DFR and the same hyperparameters as Liu et al. [25] for JTT.

We report the performance of the models via Worst-group Accuracy, or Balanced Accuracy in Table 2, which is the average of accuracies of all groups. If all groups in the test set have the same number of examples, balanced accuracy will be equivalent to average accuracy.

Our model was trained on a machine with 8 A100 GPUs.

# C Omitted Results

## C.1 Balanced Accuracies

Below we include the balanced accuracies for the experiments in Table 2.

| Method | Group Info Train / Val | CelebA-Age Balanced Accuracy | CelebA-Age Worst Group Accuracy | CelebA-Blond Balanced Accuracy | CelebA-Blond Worst Group Accuracy | Waterbirds Balanced Accuracy | Waterbirds Worst Group Accuracy | MultiNLI Balanced Accuracy | MultiNLI Worst Group Accuracy |
|---|---|---|---|---|---|---|---|---|---|
| ERM | ✗/ ✗ | 77.96 | 56.65 | 82.59 | 45.86 | 83.40 | 57.85 | 80.92 | 67.19 |
| Auto-TRAK (ours) | ✗/ ✗ | 80.05 | **75.97** | 91.01 | 83.77 | 90.36 | 81.04 | | |
| RWG [17] | ✓/ ✓ | 80.66 | **75.64** | 90.42 | 88.40 | 86.51 | 81.21 | 78.61 | 68.41 |
| SUBG [17] | ✓/ ✓ | 77.57 | 68.49 | 91.30 | 88.26 | 86.97 | 85.46 | 73.64 | 67.76 |
| GroupDRO [40] | ✓/ ✓ | 80.88 | 74.80 | 91.83 | **90.61** | 86.51 | 72.47 | 81.4 | 77.7 |
| JTT [25] | ✗/ ✓ | 68.06 | 60.95 | 92.01 | 81.61 | 85.24 | 63.61 | 78.6 | 72.6 |
| DFR [21] | ✗/ ✓✓ | 80.69 | 70.37 | 91.93 | 88.40 | 90.89 | **88.96** | 82.1 | 74.7 |
| TRAK (ours) | ✗/ ✓ | 81.05 | **75.55** | 91.08 | **90.03** | 91.46 | 87.15 | 81.54 | 75.46 |

Table 2: Balanced accuracy and worst-group accuracy on CelebA-Age, CelebA-Blond , and Waterbirds . A double checkmark (✓✓) indicates that the method uses validation group labels for model finetuning, in addition to hyperparameter tuning.

## C.2 ImageNet Accuracies

Below we included the detailed accuracies for the ImageNet experiment.

| Class (bias) | Method | Class-Level | | ImageNet-Level |
| | | Balanced Accuracy | Worst Group Accuracy | Overall Accuracy |
|---|---|---|---|---|
| Red Wolf *(Red Coat)* | ERM | 46.87 | 22.62 | 63.97 |
| | TRAK | 65.63 | **52.38** | 63.71 |
| | Auto-TRAK | 59.94 | 39.29 | 63.87 |
| Tench *(Presence of human)* | ERM | 85.10 | 78.12 | 63.97 |
| | TRAK | 90.73 | **86.88** | 63.84 |
| | Auto-TRAK | 86.67 | 80.00 | 63.97 |
| Cauliflower *(Not Cooked)* | ERM | 77.81 | 63.64 | 63.97 |
| | TRAK | 85.77 | **79.55** | 63.70 |
| | Auto-TRAK | 86.73 | 79.40 | 63.75 |
| Strawberry *(Not on a plate)* | ERM | 58.93 | 35.58 | 63.97 |
| | TRAK | 70.49 | **51.92** | 63.88 |
| | Auto-TRAK | 68.99 | 50.48 | 63.79 |

Table 3: Auto-TRAK identifies and mitigates biases in ImageNet. For four ImageNet classes, a bias was identified from inspecting the TRAK PCA directions. Then Auto-TRAK is applied in order to mitigate the bias for that class. Auto-TRAK is able to improve the worst group accuracy for the targeted class without significantly changing the overall ImageNet accuracy.

