# OpenReview forum: "Improving Subgroup Robustness via Data Selection"
_NeurIPS.cc/2024/Conference — NeurIPS 2024 poster_

### Official Review · Reviewer_yCP5 · 2024-06-27

**Soundness:** 3
**Presentation:** 3
**Contribution:** 3
**Rating:** 7
**Confidence:** 4

**Summary:**

This paper proposes a data-centric model debiasing technique to identify and remove data which harm worst-group accuracy. This method removes fewer data than standard balancing techniques and can be adapted for settings with and without group annotations. Experiments are provided on standard group robustness benchmark datasets, and the method is shown to promote bias discovery on ImageNet in the absence of group annotations.

**Strengths:**

1. The differences between the full-information, partial-information, and no-information regimes are clearly delineated, and the advantages of D3M and Auto-D3M in each setting are comprehensively discussed. In the no-information regime, which is important and yet understudied in the literature, the authors propose a novel and elegant Auto-D3M algorithm based on TRAK, which I expect to be a strong baseline for future work in this setting.
2. D3M and Auto-D3M compare favorably to other common data balancing techniques such as subsampling a class-balanced dataset, removing far fewer points while achieving better WGA performance.
3. Sections 5.2 and 6 are comprehensive and very useful for developing an intuitive understanding of the proposed group alignment scores and TRAK matrix. The ability to discover spurious correlations in complicated datasets without group annotations is likely to be useful for practitioners.
4. The explanations of each algorithm -- D3M, Auto-D3M, and TRAK -- are clear and well-written. The mathematics is well-explained and sufficiently technical without being convoluted.

**Weaknesses:**

1. In Table 1, the only comparison to previous work provided for the no-information regime is ERM, which is generally understood to be a weak baseline for group robustness tasks. Some examples of comparisons I would expect to see in this setting include MaskTune [1], uLA [2], DivDis [3], or CB-LLR [4]. Similarly, in the partial-information regime, additional comparisons may include AFR [5] or SELF [4]. (I do not expect the authors to include all these comparisons, but it would benefit to discuss the most appropriate ones).
2. In Section 6, I believe a reference and comparison to [6] is missing. Similarly to this paper, [6] uses a data-centric method to discover and mitigate spurious correlations in the ImageNet dataset.
3. Tables 1, 2, 3, and Figure 6 lack error bars. It would improve the scientific rigor of the paper to run these experiments over multiple random seeds and provide standard deviations or confidence intervals.
4. There are a couple typos and grammatical errors in the writing, e.g., on lines 482 and 484. Also, the bibtex could use an update, as some references are out of date (e.g., Kirichenko et al. -- [21] in the paper -- is listed as an ArXiv preprint but appeared at ICLR 2023).

***References***

[1] Taghanaki et al. “MaskTune: Mitigating Spurious Correlations by Forcing to Explore”. NeurIPS 2022.

[2] Tsirigotis et al. “Group Robust Classification Without Any Group Information.” NeurIPS 2023.

[3] Lee et al. “Diversify and Disambiguate: Learning From Underspecified Data.” ICLR 2023.

[4] LaBonte et al. “Towards last-layer retraining for group robustness with fewer annotations”. NeurIPS 2023.

[5] Qiu et al. “Simple and Fast Group Robustness by Automatic Feature Reweighting.” ICML 2023.

[6] Moayeri et al. “Spuriosity Rankings: Sorting Data to Measure and Mitigate Biases.” NeurIPS 2023.

**Questions:**

1. Is the hyperparameter k from D3M (number of examples to remove) the same as the hyperparameter k from TRAK (dimensionality of the gradient projection)? If not, it would be helpful to use different letters and detail how the k in TRAK is chosen.
2. Why is random initialization used for CelebA, as opposed to standard ImageNet initialization? Do the CelebA comparisons in Table 1 also use random initialization?
3. In the appendices, the tables reference proposed methods TRAK and Auto-TRAK. Is this meant to read D3M and Auto-D3M respectively?
4. While not strictly necessary, I would be curious to see a qualitative comparison of the results from Section 5.2 and Figures 3 and 4 with other data selection techniques from the robustness literature. How do the data with negative alignment scores compare with data selected via misclassification [1], disagreement [2], other influence functions [3, 4], or Shapley values [5]? Are negative alignment scores perhaps more interpretable than these other techniques?

***References***

[1] Liu et al. “Just Train Twice: Improving Group Robustness without Training Group Information.” ICML 2021.

[2] LaBonte et al. “Towards last-layer retraining for group robustness with fewer annotations”. NeurIPS 2023.

[3] Koh and Liang. “Understanding Black-box Predictions via Influence Functions.” ICML 2017.

[4] Feldman and Zhang. “What Neural Networks Memorize and Why: Discovering the Long Tail via Influence Estimation.” NeurIPS 2020.

[5] Ghorbani and Zou. “Data Shapley: Equitable valuation of data for machine learning.” ICML 2019.

**Limitations:**

While the limitations of group annotations are sufficiently discussed, I have an additional question about compute efficiency of the proposed method. The experiments were performed on 8x A100 GPUs, which is much more extensive than comparative methods (e.g., [1] only needs to train a linear classifier, which, being essentially a logistic regression, is very efficient). Is this level of compute necessary for D3M? I am mainly concerned about the TRAK subroutine of Auto-D3M, where in step (d) the covariance matrix of the projected gradients of the entire dataset is constructed and inverted T=100 times. How much wall-clock time and GPU VRAM does this step consume, and how does it scale with the hyperparameter k?

***References***

[1] Moayeri et al. “Spuriosity Rankings: Sorting Data to Measure and Mitigate Biases.” NeurIPS 2023.

---

> ### Author Rebuttal · Authors · 2024-08-07
>
> We thank the reviewer for their review, and address their questions below.
>
> **[Further comparisons to the no-information regime]**
>
> Our goal was to include the strongest baselines (to the best of our knowledge) and show that our methods perform better/comparably to them. For instance, since Auto-DDA outperforms most supervised baselines, it also outperforms weaker unsupervised (e.g., no information) baselines. However, we would be happy to include a table in the appendix in the final version of the paper with the suggested baselines.
>
> **[Additional Reference and Typos]**
>
> Thank you for the additional reference! We will include this in the final revision of our paper, and will additionally fix the typos you mentioned.
>
> **[Error bars]**
>
> We thank the reviewer for this feedback. We’ve added versions of Table 1 and Figure 6 with error bars to the global rebuttal document. Figure 2 already has a confidence region over 10 runs, but is also duplicated in the global rebuttal document.
>
> **[Hyperparameter k]**
>
> The hyperparameter k from D3M (number of examples to remove) is not the same as the k from TRAK (dimensionality of gradient projection). We will change the variable name for the final revision.
>
> **[Initialization]**
>
> Our focus in this paper is the impact of the training dataset, so we used randomly initialized models to keep the setup straightforward (though our method does not depend on the initialization scheme). We then switched to ImageNet initialization for WaterBirds specifically because we found that the base model performance is extremely poor and noisy when initialized from scratch.
>
> **[TRAK/Auto-TRAK]**
>
> We apologize for the confusion here: we used to call our method TRAK and Auto-TRAK, and then changed the names to D3M and Auto-D3M (but forgot to change the appendix), We will fix this for the final revision
>
> **[Qualitative Comparison]**
>
> We thank the reviewer for the suggestion! While we didn’t have time in this response period to experiment with other data selection techniques as suggested, we agree that this would be an interesting avenue for further investigation.
>
> **[Computation Runtime]**
>
> We appreciate the reviewer’s concern about the computational expense of our approach. We use TRAK as a subroutine in our method: a detailed analysis of the wall-clock time for TRAK can be found in Figure 1 of their paper [1]. Specifically, as noted in Appendix A.4 in their paper, the construction and inversion of the covariance matrix of projected gradients is actually quite cheap (this matches our experience): indeed, more of the runtime comes from training the models for the ensemble (TRAIN_TIME in their computation). Here, as noted in Appendix E of the TRAK paper, TRAK can be further optimized by re-using multiple checkpoints of the same model for the ensemble, further tuning the projection dimension, or even by reducing the number of models from 100 (as noted in their Appendix E.1). For simplicity’s sake, we did not explore any of these optimizations in our work, but would happy to discuss them further for the final revision.
>
> Stepping back, TRAK is simply a subroutine called by D3M and can be replaced by more performant and efficient datamodeling/data attribution methods as they emerge. The rapid pace of data attribution as a field suggests that the computational cost of this subroutine will continue to decrease with time.
>
> [1] Park, Sung Min, et al. "Trak: Attributing model behavior at scale." arXiv preprint arXiv:2303.14186 (2023).

---

> > ### Comment · Reviewer_yCP5 · 2024-08-07
> >
> > Thank you to the authors for the comprehensive response, and especially for the inclusion of additional figures with error bars. With the additional clarifications and discussion, I believe this paper will be interesting for the community and set a strong baseline for future work. Therefore, I recommend acceptance. I have raised my Soundness score to a 3 and overall rating to a 7.

---

### Official Review · Reviewer_uxzC · 2024-07-08

**Soundness:** 4
**Presentation:** 2
**Contribution:** 3
**Rating:** 7
**Confidence:** 4

**Summary:**

The paper introduces a method called Data Debiasing with Datamodels (D3M) that addresses the problem of model bias (using the worst-case loss over groups as the metric). The approach leverages a process known as datamodeling to predict model behavior based on training data influence, focusing on removing data points that contribute heavily to worst-group error. The paper illustrates D3M’s effectiveness across various datasets, showing that it can outperform both traditional model and data intervention strategies. Moreover, the method is adaptable to the setting without explicit subgroup labels.

**Strengths:**

Originality: The paper introduces an innovative approach, Data Debiasing with Datamodels (D3M), which creatively combines elements from influence functions and data-centric fairness strategies to address model bias. D3M focuses on optimizing the dataset by identifying and removing specific training instances that disproportionately skew model performance against minority subgroups. This methodological innovation brings high originality to the paper.
Quality: The authors conduct a thorough analysis across multiple datasets, effectively demonstrating how D3M enhances worst-group accuracy. The use of comparative baselines and the examination of different scenarios (including those without explicit subgroup labels) shows the robustness and reliability of D3M.
Clarity: The paper is relatively well-structured. The effective use of diagrams and necessary mathematical definitions help demonstrate the results. Moreover, case studies help readers understand the use cases of D3M.
Significance: The significance of this work is relatively substantial, addressing the issue of the subgroup biases of models. Moreover, by providing a tool that can improve model fairness without needing subgroup labels, the paper contributes to the applications where the group labels are unavailable.

**Weaknesses:**

One weakness of the method is its exclusive focus on improving worst-group accuracy without presenting results on how it might affect the overall accuracy for all groups. This raises concerns about potential trade-offs, where enhancing fairness for the worst-performing subgroup could compromise the model's general performance. Additionally, the paper does not thoroughly explore how different model configurations might influence the outcomes. Understanding how variations in model architectures, initial parameter settings, or training procedures affect the effectiveness of the method is useful for validating its robustness and adaptability to diverse scenarios. Finally, a relatively minor weakness is that the demonstration of the paper could be more organized and coherent.

**Questions:**

After improving the worst-group accuracy, does the model still maintain good overall accuracy? How does the method impact the performance across all groups? Were the results of the method tested across various model architectures to confirm its generalizability? In scenarios lacking explicit group labels, were there any experiments conducted to assess the effectiveness of the pseudo group labeling approach using the datamodel matrix in the setting or case studies in this paper?

**Limitations:**

The paper presents relatively robust results. However, I do not see it adequately address the limitations of the methods.

---

> ### Author Rebuttal · Authors · 2024-08-07
>
> We thank the reviewer for providing a thorough review of our paper. Below, we address the feedback points raised by the reviewer:
>
> **[Focus on worst-group accuracy (WGA)]**
>
> The reviewer raises the concern of evaluating only WGA without reporting overall accuracy. We note that we report balanced (i.e., average across all groups) accuracies for all of our experiments in Tables 2 and 3 in the appendix (we report WGA in the main paper due to space constraints). We will mention those results more prominently in the main paper.
>
> **[Impact of model architectures and hyperparameters on D3M performance]**
>
> We agree that investigating how model configurations change the efficacy of D3M is an promising avenue to pursue! In particular, it would be interesting to see how pre-training (e.g., on ImageNet) changes the behavior of D3M when there is a strong initial prior. For the Waterbirds dataset we do use ImageNet pre-trained models, but we don’t have a direct comparison on the same dataset between D3M’s behavior with ImageNet and randomly initialized models. We would be happy to pursue this for the final paper.
>
> **[Paper organization]**
>
> We will edit the camera-ready version of the manuscript to improve the organization of the paper where this is needed. We kindly ask the reviewer to elaborate on the paragraphs/sections which may need additional organizational work.
>
> **[Effectiveness of the pseudo-labels]**
>
> When reporting WGA for Auto-D3M (where we do not explicitly use any group labels in the method) we compute WGA using the groundtruth test group labels (e.g., in Table 1). We find that while the pseudo-labels are not exactly the same as the groundtruth group labels, they are aligned enough that performing D3M with the pseudo-labels (e.g., Auto-D3M) competitively improves worst group accuracy with respect to the groundtruth groups.
>
> In our ImageNet case study, we similarly report WGA in Figure 6 by performing manual group assignment on the 50 test images per class (e.g., for tench, we manually label for each image for “whether a human is present”). Again we find that our pseudo-labels are aligned enough with these manual labels that performing Auto-D3M improves accuracy with respect to the manually assigned groups.
>
> [1] Idrissi, Badr Youbi, et al. "Simple data balancing achieves competitive worst-group-accuracy." Conference on Causal Learning and Reasoning. PMLR, 2022.

---

> > ### Comment · Reviewer_uxzC · 2024-08-10
> > **Reply**
> >
> > I appreciate that the authors refer to Tables 2 and 3. It would be acceptable to keep them in the Appendix, but it would be better to highlight this with one or two sentences in the main body of the paper. Regarding the organization of the paper, I feel that Section 4 could benefit from more focused editing, as it encompasses discussions from multiple perspectives. It might be helpful to provide a clearer outline at the beginning of the section, or perhaps divide it into subsections with a summary at the start of each. For different model configurations, I believe it would be a significant weakness if comparisons are not conducted. I hope this will be addressed in the final version of the paper. I look forward to the final draft.

---

### Official Review · Reviewer_QjGu · 2024-07-12

**Soundness:** 2
**Presentation:** 3
**Contribution:** 3
**Rating:** 5
**Confidence:** 5

**Summary:**

This paper introduces a new data debiasing technique called Debiasing with Data Attribution (DDA). DDA utilizes data modelling framework to identify and eliminate training examples that negatively impact the accuracy of the worst-performing groups. Additionally, the paper presents AUTO-DDA, an extension of DDA that can identify biases even without prior knowledge of group information. The proposed methods are validated through experiments on various datasets such as CelebA-Age, CelebA-Blond, Waterbirds and MultiNLI.

**Strengths:**

1. The proposed approach is simple and effectively improves the performance on real-wolrd datasets such as ImageNet.
2. The paper is presented well and easy to follow.

**Weaknesses:**

1. The performance of ImageNet is only reported on selected classes. How are the classes selected for evaluation? Is it based on the amount of bias present in the classes?
2. I am unsure if the proposed approach is effective when the majority of the data consists of bias-aligned points. For example, if there are only a few conflicting points and the rest are bias-aligned, how will the data be removed? I doubt the approach would still be useful for debiasing since a large part of the data is still going to be majorly biased.  Even if the authors claim that the majority of the bias aligned points will be removed, I believe the model would still overfit to the data since the final dataset would be extremely small. Analyzing the performance of the approach with varying numbers of bias-conflicting points (1%, 5%, 10% of CMNIST(10 class classification)) in the dataset would be beneficial to understand this scenario. This experiment would provide insights into how well the approach scales to real-world scenarios where the degree of bias is significantly high.

**Questions:**

Please refer to the questions in the weakness section.

**Limitations:**

Limitations are discussed.

---

> ### Author Rebuttal · Authors · 2024-08-07
>
> We would like to thank the reviewer for their feedback on our work. Below, we have address the concerns raised by the reviewer:
>
> **[Selection of specific classes chosen for ImageNet experiments]**
>
> We selected classes that previous work found had biases in the ImageNet dataset. Specifically biases in the classes Red Wolf, Tench, Cauliflower, and Strawberry were studied in [1] and biases in the classes Snorkel, Howler Monkey, and Dog Sled were studied in Hard ImageNet [2].
>
> **[Debiasing datasets with few bias-conflicting examples]**
>
> As the reviewer points out, if basically all the training data points are bias-aligned points (and there are only a couple of examples that are not aligned) a dataset selection method may result in a very small training dataset which risks the model overfitting. This is a misalignment of objectives: here we are optimizing for worst group accuracy. In such a case, the worst group accuracy of the ERM model would be close to zero, since we entirely rely on the bias and thus completely fail on the non-bias aligned points. Reducing to a small dataset through dataset selection might degrade the overall accuracy, but will still likely improve the worst group accuracy.
>
> However, even in this case, D3M is a better approach than dataset balancing, since we are more sample efficient when removing data points (removing the “worst offenders”) as demonstrated in Figure 2. Moreover, both Celeba-Blond and Waterbirds are highly bias-aligned datasets, where the smallest group constitutes a very small number of examples. As we show in Table 1, D3M is still effective in these scenarios.
>
>
> [1] Jain, Saachi, et al. "Distilling model failures as directions in latent space." arXiv preprint arXiv:2206.14754 (2022).
>
> [2] Moayeri, Mazda, Sahil Singla, and Soheil Feizi. "Hard imagenet: Segmentations for objects with strong spurious cues." Advances in Neural Information Processing Systems 35 (2022): 10068-10077.

---

> > ### Comment · Reviewer_QjGu · 2024-08-08
> > **Reply**
> >
> > **[Selection of specific classes chosen for ImageNet experiments]** Thanks for clarifying this.
> >
> > **[Debiasing datasets with few bias-conflicting examples]**
> > I request the authors to please validate the claim emperically, by training a simple 3 layer MLP using D3M for various percental of bias conflicting samples (0.05%, 1% and 5%) on CMNIST dataset where the proportion of bias conflicitng samples can be easily controlled as suggested in my intial review. Also ablating on various percentage of bias conflicing samples helps understand the sensitivity of the proposed methods on number of bias conflicitng samples about which im concerned about.

---

> > > ### Comment · Reviewer_QjGu · 2024-08-13
> > > **Reply**
> > >
> > > My concerns on performance on **Debiasing datasets with few bias-conflicting examples** has not been addressed, hence Iam lowering my score.

---

> ### Author Response · Authors · 2024-08-13
> **Experiment is incoming!**
>
> HI reviewer QjGu,
>
> We really appreciate your suggested experiment, and have been working for the last few days straight on implementing it and getting the results (we are somewhat compute-constrained at the moment)! We really appreciate your patience and should have results by the end of the day today.
>
> - Authors

---

> > ### Author Response · Authors · 2024-08-14
> > **Experiment with few bias-conflicting examples**
> >
> > Thank you for your patience! We've run the experiment you requested.
> >
> > ## Setup
> >
> > For the MNIST data
> > - Let $\hat{y}$ be the binary signal according to the digit: (0 if $<5$ and 1 if $\geq 5$). The true label $y$ has a 90% chance of being $\hat{y}$ and 10% chance of being $1-\hat{y}$
> > - Let $y_{col} = y$ with $p_{corr}$ probability and $1-y$ otherwise. If $y_{col}$ is 1, color the image red, otherwise if 0 color the image green.
> >
> > Thus the color ($y_{col}$) aligns with the true label ($y$) with $p_{corr}$ probability and the digit $\hat{y}$ with 0.9 probability. If $p_{corr} > 0.9$, the ERM model relies more heavily on the color than the digit.
> >
> > We take val as 10% of the training split of MNIST. We consider three splits for train where there are very few bias conflicting examples.
> > - $p_{corr} = 0.95$
> > - $p_{corr} = 0.99$
> > - $p_{corr} = 0.995$
> >
> > And our test set has $p_{corr} = 0.1$ (so the bias alignment is reversed).
> >
> > Here, the groups are the pair of $y, y_{col}$ (e.g., the final class and the color of the image.)
> >
> > ## Results
> >
> > | $p_{corr}$    | ERM WGA | D3M WGA | Balancing WGA (oracle) | ERM Avg. Acc | D3M Avg. Acc | Balancing Avg. Acc (oracle) | Frac Examples Removed by D3M |
> > |:--------:|:-------:|:-------:|:-------:|:-------:|:-------:|:-------:| :-------:|
> > | $p_{corr} = 0.95$  |  $20.52 \pm 0.85$  | $44.48 \pm 0.72$  | $78.66 \pm 1.82$ | $29.00 \pm 0.37$   |  $89.79 \pm 0.06$ | $82.78 \pm 0.44$ | 0.73  |
> > | $p_{corr} = 0.99$  |  $1.95 \pm 0.25$  | $55.15 \pm 3.53$ | $69.98 \pm 3.21$ | $12.00 \pm 0.22$   |  $87.99 \pm 0.065$ | $75.86 \pm 0.66$ | 0.80 |
> > | $p_{corr} = 0.995$  |  $0.62 \pm 0.13$  |  $42.24 \pm 2.05$  | $26.16 \pm 17.62$ | $10.62 \pm 0.11$   | $87.31 \pm 0.21$ | $60.19 \pm 6.55$ | 0.83 |
> >
> > **ERM vs. D3M**: We find that even at these more extreme scenarios of bias (where the vast majority of examples are removed), we can significantly improve both WGA and average test accuracy of the model using D3M over using ERM. Our performance does not significantly degrade even at $p_{corr} = 0.995$ where there are very few bias conflicting examples.
> >
> > **Number of examples removed**: Also, note that to balance the dataset, more than 90% of examples would need to be removed. Since our method is more sample efficient than balancing, we can remove a smaller fraction of the examples. Our heuristic for choosing $k$ (the number of examples to remove) *overshoots* the best number. In particular, the previously majority classes are now the worst performing groups in all cases: we can thus improve the accuracies further by removing *even fewer* examples (e.g., by searching for $k$ on the validation set).
> >
> > **Comparison to balancing**: We further compare to balancing the dataset.  We note that this method requires *training group annotations*, while our method does not. In this simplified setting, most of the examples are equally harmful for WGA, so naive balancing can do well. Here balancing the dataset improves WGA over D3M for $p_{corr} = 0.95, 0.99$. However, balancing more significantly degrades overall accuracy as there are very few points remaining in the dataset. For $p_{corr}  = 0.995$, balancing the dataset requires such a small overall training dataset that both WGA and overall accuracy are harshly degraded compared to D3M.
> >
> > We would be happy to include this result within our paper. We apologize for the delay: we worked hard over the weekend to get these results in (but were delayed due to some cluster issues). We would appreciate if the reviewer would revisit their decision to lower their score, given this experiment.

---

### Official Review · Reviewer_FTVd · 2024-07-13

**Soundness:** 3
**Presentation:** 2
**Contribution:** 3
**Rating:** 5
**Confidence:** 2

**Summary:**

The paper proposes Data Debiasing with Datamodels (D3M), a method to improve machine learning model performance on underrepresented subgroups by removing specific training examples that cause failures. Unlike traditional balancing methods, D3M efficiently debiases classifiers without needing group annotations, significant dataset reductions or additional hyperparameter tuning.

**Strengths:**

- Significance: This work effectively identifies and removes training samples to improve the worst-group accuracy. As demonstrated by the experiments, this method outperforms both standard model-based and data-based approaches.
- Comprehensive Datasets: A wide range of datasets is used for image and text classification tasks, with corresponding benchmarks evaluated against existing methods as listed in Appendix B, "Details of Experiments."

**Weaknesses:**

- Writing and Format: the presentation of the paper needs readability improvement:
  - Redundant section start: Line 81
  - Excessive parenthetical comments and irregular format: Lines 20, 25, 28, 32, 67-71, 83, etc.

- In addition to isolating problematic training data, experiments should be conducted to assess the impact on the necessity of further hyperparameter tuning and to strengthen the case for the effectiveness of the proposed method.

**Questions:**

N/A

**Limitations:**

See weaknesses

---

> ### Comment · Area_Chair_pNCY · 2024-08-08
> **Please read rebuttal and provide more substantive comments**
>
> Dear reviewer FTVd,
>
> Your review appears to mostly mention formatting issues. Please read the authors' response to other reviews and provide comments regarding the content of the paper, if you have any.
>
> Thanks,
> AC

---

> > ### Comment · Reviewer_FTVd · 2024-08-11
> >
> > I have read the other reviews and the authors' responses. I look forward to the final draft of the paper, as the authors will be editing the camera-ready version to improve its organization. Regarding the content, I appreciate the insights of the other reviewers and would like to increase my score to 4.

---

### Author Rebuttal · Authors · 2024-08-07

Thank you for your reviews! We respond to the questions of each reviewer individually below. We additionally include results with error bars in the attached PDF as requested by Reviewer yCP5.

---

### Decision · Program_Chairs · 2024-09-25

**Decision:**

Accept (poster)

**Comment:**

This paper proposes a new approach to group robustness by identifying and removing specific examples that drive poor performance on minority groups (thus called "dataset debiasing"). The reviewers are broadly enthusiastic about the contribution. While reviewers initially raised concerns, e.g. along the lines of evaluation of selected classes/groups on datasets and comparison with only a few benchmarks, most of these were resolved in the rebuttal and discussion phase.

I encourage the authors to incorporate all of the new experiments/empirical results that were provided during the rebuttal and discussion phase in the camera-ready version, as these were instrumental to the paper's acceptance. I also encourage the authors to take seriously all of the suggestions that were made for formatting improvement.